# Southern African Origin of HTLV-1 in Romania

**Olivier Cassar**[1☯]*, **Ambroise Marçais**[2☯], **Olivier Hermine**[2], **Emilie Deruelle**[2],
**Giovanni Begliomini**[1], **Alexandru Bardas**[3], **Horia Bumbea**[4], **Andrei Colita**[5], **Daniel Coriu**[3],
**Viola Maria Popov**[6], **Alina Tanase**[3], **Philippe Vicente Afonso**[1], **Antoine Gessain**[1]*

**1** Institut Pasteur, Université Paris Cité, CNRS UMR 3569, Unité d'Épidémiologie et Physiopathologie des Virus Oncogènes, Département de Virologie, Paris, France, **2** Hôpital Necker-Enfants Malades, Service d'Hématologie, Paris, France, **3** Fundeni Clinical Institute, University of Medicine and Pharmacy Carol Davila, Bucharest, Romania, **4** Emergency University Hospital, University of Medicine and Pharmacy Carol Davila, Bucharest, Romania, **5** Coltea Hospital, University of Medicine and Pharmacy Carol Davila, Department of Hematology, Bucharest, Romania, **6** Colentina Clinical Hospital, Hematology 1 Department, Bucharest, Romania

☯ These authors contributed equally to this work.
* olivier.cassar@pasteur.fr (OC); antoine.gessain@pasteur.fr (AG)

**Data Availability Statement:** The data that support the findings of this study are available from GenBank (Accession numbers: OR852771 to OR852781 and OR825456 to OR825465).

## Abstract

In Europe, most HTLV-1-infected individuals originate from highly endemic regions such as West Indies, sub-Saharan Africa, and South America. The only genuine endemic region for HTLV-1 in Europe is Romania where ATL series have been reported among Romanian patients. Our objective is to better understand the origin of this endemic focus based on a study of the genetic diversity of HTLV-1 in Romanians. DNA was obtained from PBMCs/buffy coats of 11 unrelated HTLV-1-infected individuals of Romanian origin. They include 9 ATL cases and 2 asymptomatic carriers. LTR sequences were obtained for all specimens. Complete genomic HTLV-1 sequences were obtained using four PCR series on 10 specimens. Phylogenetic trees were generated from multiple alignments using HTLV-1 prototypic sequences and the new generated sequences. Most of the complete LTR sequences (756-bp) showed low nucleotide diversity, ranging from 0% to 0.8% difference, and were closely related (less than 0.8% divergence) to the only previously characterized Romanian strain, RKI2. One strain, ROU7, diverged slightly (1.5% on average) from the others. Phylogenetic analyses both on partial LTR and the complete genome demonstrate that the 11 sequences belong to the HTLV-1a cosmopolitan genotype and 10 of them belong to the previously denominated a-TC Mozambique–Southern Africa A subgroup. In this study, we demonstrated that the HTLV-1 present in Romania most probably originated in Southern Africa. As most Romanian HTLV-1 strains are very closely related, we can assume that HTLV-1 has been introduced into the Romanian population recently. Further studies are ongoing to decipher the routes of arrival and dissemination of these HTLV-1 strains, and to date the emergence of this endemic focus in Central Europe.

## Author summary

In Europe, HTLV-1 infections mainly originate in regions such as the West Indies, sub-Saharan Africa and South America. However, Romania has been identified as a true

**Funding:** o AG received funding from the French Government's Investissement d'Avenir Program, Laboratoire d'Excellence "Integrative Biology of Emerging Infectious Diseases" (Grant no. ANR-10-LABX-62-IBEID), the Centre National de la Recherche Scientifique (CNRS-UMR 3569) and from the Institut Pasteur, Paris, France. The funders had no role in study design, data collection and analysis, decision to publish, or preparation of the manuscript. None of the authors received a salary from the funders.

**Competing interests:** The authors have declared that no competing interests exist.

endemic area, due to reported cases of ATL among Romanian patients. To understand this situation, we analyzed the genetic diversity of HTLV-1 in Romanians and collected DNA from 11 infected individuals, most of them ATL cases. The results showed a close genetic relationship between most of the sequences, closely matching the previously identified Romanian RKI2 strain. However, one strain, ROU7, showed a slight divergence. Phylogenetic analysis positioned these sequences in the cosmopolitan HTLV-1a genotype, belonging mainly to the a-TC Mozambique-Southern Africa subgroup. This suggests a probable origin of HTLV-1 in Romania from South Africa, perhaps recently introduced. Further studies aim to elucidate the transmission routes and emergence of this HTLV-1 endemicity in Central Europe.

## Introduction

Human T-cell lymphotropic virus type 1 (HTLV-1) is a human oncoretrovirus infecting at least 5 to 10 million individuals worldwide. It has major public health implications, mainly due to its causal association with severe diseases such as adult T-cell leukemia/lymphoma (ATL) and HTLV-1-associated myelopathy/tropical spastic paraparesis (HAM/TSP) [1]. Based on genetic variability, seven main HTLV-1 molecular genotypes (a to g) have been described in certain geographic areas and ethnic groups [2,3]. Although the virus is highly endemic in Southern Japan, sub-Saharan Africa, the Caribbean and Australo-Melanesia, its prevalence varies significantly from one population to another [2]. Moreover, HTLV-1 demonstrates remarkable genetic stability, with the observed polymorphism among viral strains correlating with the geographic origin of infected individuals. This limited genetic drift offers a molecular avenue for tracking viral transmission and the migrations of historical infected populations. In Europe, such as France, the United Kingdom or Spain, HTLV-1 is rare, except in people who have immigrated from highly endemic regions such as sub-Saharan Africa, the West Indies and South America, respectively [1,2].

Over the past 30 years, several studies indicate that Romania, a Central European country is a HTLV-1 endemic area in Europe [4–10]. This is based, on the one hand, on studies of blood donors (mandatory screening since 1999) with a seroprevalence of up to 0.053 in first time blood donors (ten times higher than in France and the UK) [11] and, on the other hand, on reports of ATL cases, mainly sporadic ones or small series [5–8], but also recently a large series of 56 patients [9]. However, no specific studies have been carried out to determine the origin of this virus. Therefore, to get new insights into this origin, we decided to study the genetic diversity of HTLV-1 present in Romanians.

## Material and methods

### Ethics statement

Formal written consent was obtained from participants in this study, which was approved by the Human Protection Committee Ile de France II (Comité de Protection des Personnes—CPP IdF II) registred as an Institutional Review Board (IRB registration number: 000001072) and the French Data Protection Authority (Commission Nationale de l'Informatique et des Libertés—CNIL registration number: 1692254).

**Table 1. Characteristics of HTLV-1 patients from Romania, including 9 cases with Adult T-cell leukemia (ATL).**

| ID | Birth country | Age at diagnosis (y) | Gender | ATL clinical status | HIV status | PVL (%)[&] |
|---|---|---|---|---|---|---|
| ROU1[¶] | Romania | 27 | F | Chronic | Negative | 40 |
| ROU2 | Romania | 32 | F | Acute | Negative | >50 |
| ROU3 | Romania | 39 | F | Acute | Negative | >50 |
| ROU4 | Romania | 26 | M | Acute | Negative | 31 |
| ROU5 | Romania | 42 | F | Chronic | Unknown | 51 |
| ROU6 | Romania | 48 | M | Chronic | Negative | 40 |
| ROU7[#] | Romania | 37 | F | Carrier | Negative | <1 |
| ROU8[¶] | Romania | 30 | F | Carrier | Negative | 26 |
| ROU9 | Romania | 53 | F | Acute | Unknown | NA |
| PH523 | Romania | 77 | F | Lymphoma | Unknown | NA |
| PH630[#] | Romania | 17 | F | Lymphoma | Unknown | NA |

ATL: adult T-cell leukemia/lymphoma; NA: Not available.

[&]Proviral load (PVL) expressed as a percentage of infected PBMCs.

[#]HTLV-1 seropositive mother.

[¶]Blood transfusion history

## Studied individuals and data collection

We included eleven HTLV-1 unrelated infected individuals living in Romania, mostly suffering from ATL (Table 1). They were referred for consultation and sometimes therapeutic treatment, mainly to the hematology department of Necker-Enfants Malades hospital (APHP) in Paris. They were all born in Romania, most of them living in Bucarest and selfidentify as Caucasian.

## HIV and HBV screening and blood transfusion history

All but 4 patients (ROU5, ROU9, PH523 and PH630) were tested for HIV. HBV infection status was known for only 5 individuals (ROU1, ROU2, ROU3, ROU7 and ROU9). Two individuals have reported transfusion events that took place in Romania during infancy (ROU1 and ROU8).

## HTLV-1 amplification and sequencing

High molecular weight DNA samples obtained from peripheral blood buffy coats were subjected to four series of PCR (F1-F4) using LTR-*gag* (F1), *pro-pol* (F2), *pol-env* (F3), and Px-LTR (F4) primers, which were designed from highly conserved regions for the major HTLV-1 subtypes. The LTR-*gag* primers are the following: Enh280: 5′-TGACGACAACCCCTCACC TCAA-3′ and R2380: 5′-GTCCGGAAAGGGAGGCGTATTAG-3′ corresponding to nucleotides 258 to 279 and 2,377 to 2,399 respectively of the prototype ATK-1 sequence (Genbank: J02029). The *pro-pol* primers are the following: F2279: 5′-GGAGCAGACATGACAGTCC TTCC-3′ (nt 2,254 to 2,276) and R5005: 5′-GGCGGCTATTAAGACCAGGAAGC-3′ (nt 5,002 to 5,024). The *pol-env* primers are the following: F4583: 5′-CAGGAGCCATCTCAGCTAC CC-3′ (nt 4,560 to 4,580) and Env22: 5'-GGCGAGGTG GAGTCCTTGGAGGC-3' (nt 60752 to 6,774). Finally, the Px-LTR primers are the following: F4bisfor: 5′- CCCTCCTTTACCC ATCGTTAG -3′ (nt 5,941 to 5,961) and F4bisRev: 5′- CGCAGAACAGAAAACGAAAC-3′ (nt 8,867 to 8,886). The size of the different generated amplicons are the following: LTR-*gag*, 2,136-bp; *pro-pol*, 2,769-bp; *pol-env*, 2,078-bp and Px-LTR, 2,905-bp.

A PIKO thermocycler (Ozyme, Saint Quentin-en-Yvelines, France) was used with the following amplification conditions for the first 3 fragments (F1-F3): 98˚C, 30 s; 40× (98˚C, 5 s; 67˚C, 5 s; 72˚C, 1 mn); 72˚C, 1 mn and then for the last fragment (F4): 98˚C, 30 s; 40× (98˚C, 5 s; 61˚C, 5 s; 72˚C, 1 mn); 72˚C, 1 mn. Reaction tubes were prepared in a dedicated room outside the laboratory, with a final volume of 50 µl (DNA matrix, 250 ng; dNTP mix (Roche, Basel, Switzerland), 40 µM; 5×Phire reaction buffer which contains 1.5 mM MgCl2 at final reaction concentration (Life Technologies, Courtaboeuf, France), 5 µl; Phire hot start DNA polymerase (Life Technologies, Courtaboeuf, France), 2 U and 0.5 µM of each oligonucleotide primer (Eurofins Genomics, Germany). Five µl of amplified DNA was size fractionated by 1.5% agarose gel electrophoresis, and the PCR products (45 µl) were sent for purification and sequencing reactions to the Eurofins Genomics Company (Eurofins Genomics, Cologne, Germany).

The 4 fragments obtained were then sequenced using 18 pairs of primers [12] and the ClustalW algorithm (MacVector 18.6.1 software, Oxford Molecular) was implemented to align forward and reverse sequences of each segment to derive a consensus sequence of the full LTR (756-bp) region, and colinearized *gag-pol-env-tax* (7,567-bp) genes.

### Phylogenetic analyses

For the LTR and concatenated gag-pol-env-tax genes analyses, sequences were aligned using the DAMBE program (version 4.2.13). The final alignment was submitted to the Modeltest program (version 3.6) and the best model was selected according to the Akaike information criterion. This was then applied to phylogenetic analyses using the PAUP program (version 4.0b10) to infer trees according to Neighbor-Joining (NJ) method. The tree architecture was later validated by the Maximum Likelihood method, using SeaView (version 5); the robustness of the clades was estimated by approximate likelihood-ratio test (aLRT).

### LTR analyses

Phylogenetic comparison was performed on 485-nucleotide-long LTR alignment of isolates, including the 11 sequences generated in this study (in bold red) and 70 reference strains. The analysis was only carried out on 485-bp because the majority of sequences available in the LTR are not complete and this size, although reduced compared to the complete LTR (756-bp), made it possible to include a significant number of sequences from the different subgroups known in the transcontinental genotype. Two Transcontinental Japanese LTR sequences were used as outgroup (a-JPN). The phylogenetic tree was derived by the neighbor-joining method using the GTR model (gamma = 0.2884). Horizontal branch lengths are drawn to scale, with the bar indicating 0.005 nucleotide replacement per site. Numbers on each node indicate the support probability (approximate likelihood-ratio test, aLRT, calculated using the SeaView 5 software) for each cluster (Fig 1). Next to each sequence, three letters symbolize the country of origin of the infected individual (mostly IOC country codes): MOZ-Mozambic, RSA-Republic of South Africa, SWZ-Eswatini, IRN-Iran, KWT-Kuwait, SEN-Senegal, DRC- Democratic Republic of the Congo, UGA-Uganda, CIV-Côte d'Ivoire, ROU-Romania, TGO-Togo, AGO-Angola, NIG-Nigeria, JPN-Japan. The topology of the phylogenetic tree was validated by Maximum Likelihood method. The groups of interest are colored as follows: grey, blue, purple, orange, green, yellow belong to Transcontinental (TC) HTLV-1a genotype, Mozambic Southern Africa (B), Mozambic-Middle East, Mozambic, Mozambic Southern Africa (A) and Japanese (a-JPN), respectively. The GenBank accession nos OR852771 to OR852781 correspond to the 11 new LTR sequences obtained from Romanian ATL patients.

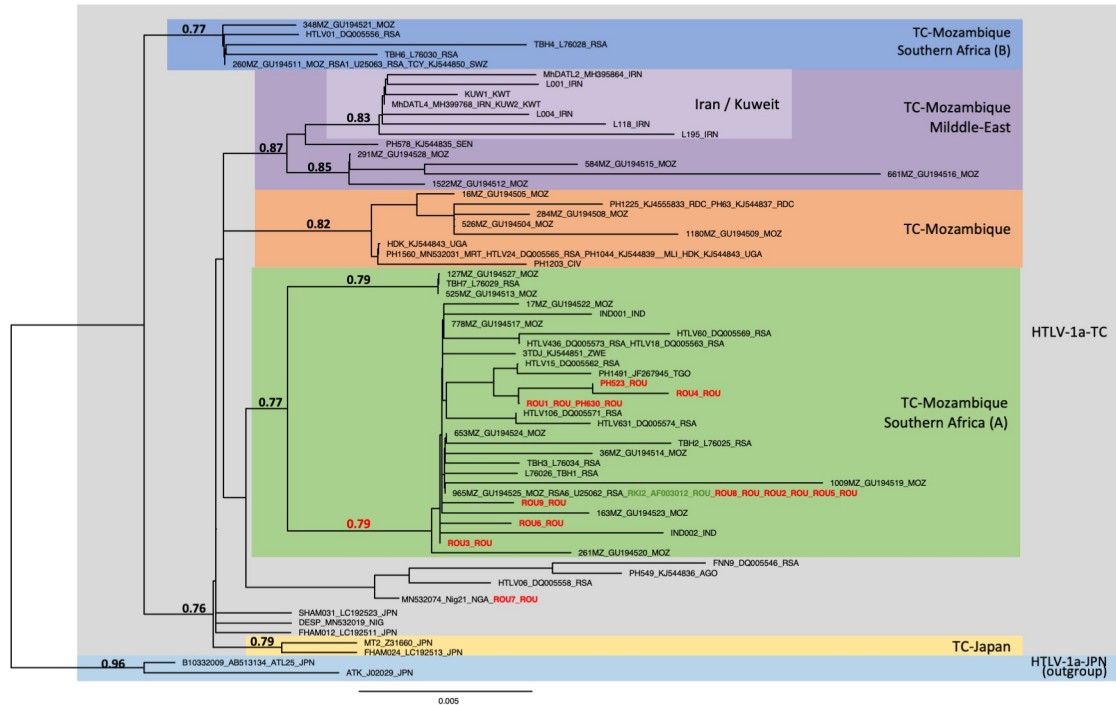

**Fig 1. Phylogenetic tree generated with neighbor-joining (NJ) method on a 485-bp fragment of the LTR region for 81 HTLV-1 available sequences including the 11 generated in this work (in bold red type).** The numbers at some nodes of the tree correspond to the probability of each monophyletic groupe (a-LRT). The branch lengths are drawn to scale with bar indicating 0.005-nucleotide replacement by site. The ATK-1 and ATL-25 strains were used as outgroup.

### Gag-pol-env-tax genes analyses

Phylogenetic analysis was derived from Neighbor-Joining method using the GTR model (gamma = 0.4815) and was confirmed using the Maximum Likelihood method. Phylogenetic tree was obtained from concatenated fragments of the 6,366-nucleotide-long *gag-pol-env-tax* genes, generated from 46 available complete HTLV-1 genomes comprising the sequences of genotypes a, b and c and the ten sequences generated in this work (in red bold type). Five complete Australo-Melanesian HTLV-1c sequences were used as outgroup. The topology of the tree was identical when the different genes were analyzed separately. Horizontal branch lengths are drawn to scale, with the bar indicating 0.01 nucleotide replacement per site. Numbers on each node indicate the aLRT for the supported cluster (Fig 2). The GenBank accession nos OR825456 to OR825465 correspond to the 10 new full HTLV-1 sequences obtained from Romanian ATL patients.

### Results

The 11 HTLV-1 infected individuals from Romania included 9 women (mean age 39.3, range 17–77) and 2 men (mean age 37 years) (Table 1). Of the eleven patients, four had acute ATL, three had chronic ATL and two had ATL lymphoma. In addition, two were asymptomatic carriers. The proviral load—expressed as a percentage of infected peripheral blood mononuclear cells (PBMCs)—ranged from less than 1% to over 50%. We were able to test for the presence of HIV antibodies in 6 people, all of whom were negative (ROU1, ROU2, ROU3, ROU4, ROU7 and ROU8) (Table 1). Regarding hepatitis B virus status, only 5 individuals were tested: one

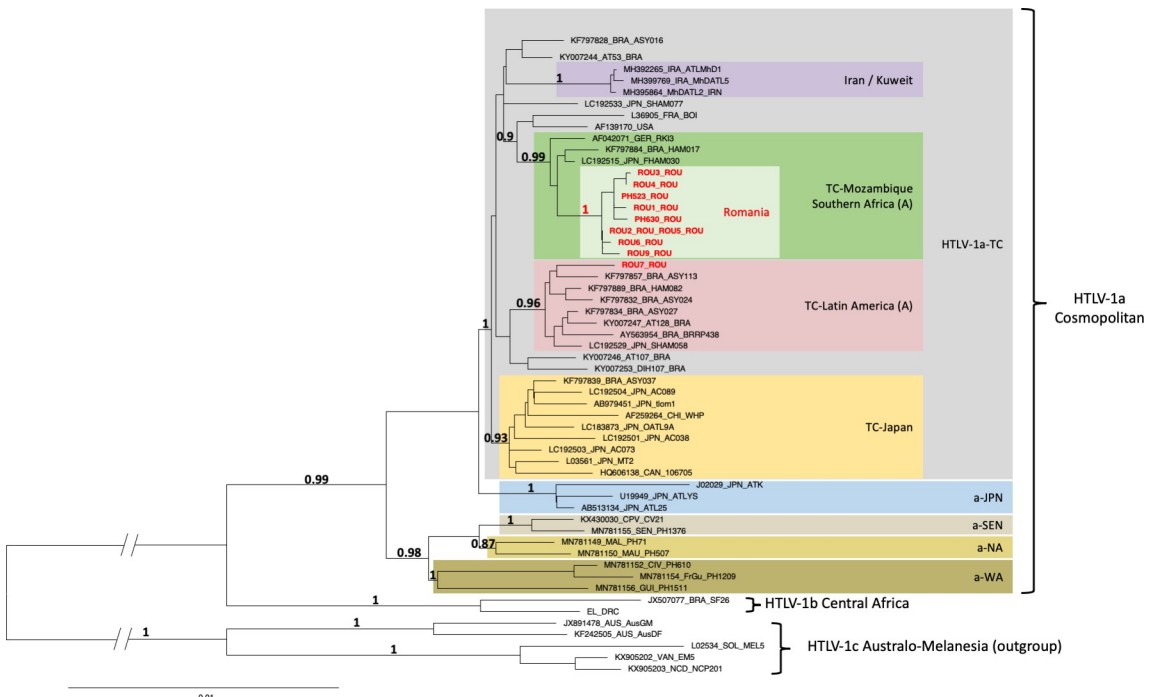

**Fig 2. Phylogenetic tree generated with neighbor-joining (NJ) method on a 6,366-bp fragment of the HTLV-1 *Gag-Pol-Env-Tax* concatenated genes for 56 available sequences including the 10 generated in this work (in bold red type).** The numbers at some nodes of the tree correspond to the probability of each monophyletic groupe (a-LRT). The branch lengths are drawn to scale with bar indicating 0.01-nucleotide replacement by site. Five HTLV-1c Australo-Melanesian strains were used as outgroup.

was vaccinated (ROU1), one had antibodies indicating past infection (ROU2), two were negative (ROU3 and ROU7) and the last one was HbS carrier, indicating chronic infection (ROU9). Two patients had a history of blood transfusion performed in Romania at an early age: ROU1 was transfused during the neonatal period, and ROU8 had multiple transfusions during childhood for acute leukemia (5 years old).

## Long Terminal Repeat (LTR) region analysis

The complete LTR sequence was obtained for the 11 samples from PCR fragments F1 and F4. Three sequences were identical (ROU2, ROU5 and ROU8). Most sequences displayed a low nucleotide diversity, i.e. less than 0.8% difference, and were closely related (less than 0.8% divergence) to the only previously characterized Romanian strain, RKI2 [13]. One strain, ROU7, diverged slightly (1.5% on average) from the others.

Phylogenetic analysis was performed on 485-bp-long LTR alignment of isolates, including the 11 sequences generated in this study (in bold red) and 70 reference strains which represent the different subtypes within HTLV-1 transcontinental (a-TC) strains (Fig 1). These subgroups were defined and named according to previous studies based on analysis of the LTR region of strains from very different geographical origins [3,14]. The analysis shows that the 11 new Romanian strains belong to the transcontinental HTLV-1a subgroup (a-TC). In addition, all but one (ROU7) belong to the a-TC Mozambique–Southern Africa A subgroup, in which Romanian RKI2 strain is also present (Fig 1). ROU7 is positioned outside this group.

The 11 new characterized sequences from Romania do not belong to the TC-Japan or TC-Middle East subgroups.

### Colinearized gag-pol-env-tax genes fragment analysis

Due to limited DNA availability for sample ROU8, we were only able to generate the complete HTLV-1 proviral sequence for only 10 out of these 11 samples. Alignment of a 7523 nt long segment of the 10 genomes–the complete genome without LTRs–shows that, as with LTR, 9 strains were very close to each other: 2 are identical (ROU2 and ROU5) and the maximal nucleotide divergence is 0.2%. One strain (ROU7) was more distant with a divergence of around 1%.

Analysis of the sequences showed that the open reading frames (ORFs) were conserved, with the exception of strain PH630, which presents a stop codon in the *pol* gene.

Phylogenetic tree was obtained from concatenated fragments of the 6366-nucleotide-long *Gag-Pol-Env-Tax* genes, generated from 46 available complete HTLV-1 genomes comprising the sequences of genotypes a, b and c and the ten new sequences generated in this work (in red bold type) (Fig 2). This phylogenetic analysis confirms that the 10 new Romanian HTLV-1 strains all belong to the HTLV-1 a-TC subgroup. Notably, 9 of the 10 Romanian strains (sample ROU7 being the exception) form a specific "Romania" clade, within the highly supported a-TC Mozambic-Southern African A subgroup (Fig 2). The topology of the tree was identical when the different genes were analyzed separately.

## Discussion

In this molecular epidemiological study, we demonstrated that the HTLV-1 present in Romanians living in Romania most probably originated in Southern Africa. As most Romanian HTLV-1 strains are very closely related, and sometimes identical, we can assume that HTLV-1 has been introduced into the Romanian population fairly recently with a limited number of strains, which leads to a founder viral effect. Furthermore, analysis of the different HTLV-1 genes separately generated a similar topology of the phylogenetic tree, consistent with the fact that mutations are found scattered throughout the genome. This suggests that these viruses were not generated by recombination [15].

Concerning the presence of a stop codon in the *pol* gene of the proviral strain of sample PH630, it has been suggested that ATL cells are clonally selected notably by acquiring a number of genetic alterations in viral genes (*gag*, *pol*, *tax*) that favor escape from the host immune system [16].

ATL develops preferentially in people infected with HTLV-1 in childhood, and rarely occurs in those infected in adulthood [17]. It is therefore likely that the ATL patients included in this study were infected at an early age. Interestingly, two patients had HTLV-1-seropositive mothers (PH630 and ROU7), suggesting possible mother-to-child transmission. The transfusion of two patients (ROU1 and ROU8) at an early age in Romania may explain their HTLV-1 positivity.

On the basis of the data available at present, we cannot demonstrate, or even suggest, how this virus entered Romania and then spread there. It is tempting to think that there is some similarity between the situation regarding the spread of HTLV-1 and that which existed a few decades ago in Romania for the epidemics of HIV and hepatitis B, particularly in children [18,19]. However, only solid, in depth epidemiological studies specifically seeking for risk factors for HTLV-1 acquisition in the Romanian population and the families of infected individuals, combined with historical data, will be needed to try to better understand the origin of the current situation. The fact that cases of ATL appeared clinically several decades ago in patients who were then on average 40 years old (this report, 5, 9) suggests occurences of HTLV-1 infection in Romania dating back to nearly 60 years ago. It should be noted that this epidemiological work will certainly be difficult to carry out due to the retrospective aspect of these studies,

which therefore concern facts that are already old, associated with public health decisions taken at the time. The recent discovery that Moldavia, a small border country to the east of Romania, also appears to be an area of high HTLV-1 endemicity, based on blood donors studies, should also encourage local public health authorities to carry out this type of epidemiological study [20].

Above all, it is essential to pursue surveillance and research efforts to limit the spread of this virus within the Romanian population.

## Acknowledgments

We acknowledge the assistance of Drs Frenoy N. from Paul Brousse hospital (Villejuif, France) and Baumelou E. from Foch hospital (Suresnes, France), where some clinical specimens were collected.

## Author Contributions

**Conceptualization:** Olivier Cassar, Ambroise Marçais, Olivier Hermine, Philippe Vicente Afonso, Antoine Gessain.

**Data curation:** Olivier Hermine, Emilie Deruelle.

**Formal analysis:** Olivier Cassar, Giovanni Begliomini, Philippe Vicente Afonso, Antoine Gessain.

**Funding acquisition:** Antoine Gessain.

**Investigation:** Olivier Cassar, Ambroise Marçais, Olivier Hermine, Emilie Deruelle, Alexandru Bardas, Horia Bumbea, Andrei Colita, Daniel Coriu, Viola Maria Popov, Alina Tanase, Antoine Gessain.

**Methodology:** Olivier Cassar, Olivier Hermine, Philippe Vicente Afonso, Antoine Gessain.

**Project administration:** Olivier Hermine.

**Resources:** Ambroise Marçais, Emilie Deruelle.

**Supervision:** Olivier Hermine, Antoine Gessain.

**Validation:** Olivier Cassar, Olivier Hermine, Antoine Gessain.

**Writing – original draft:** Olivier Cassar, Philippe Vicente Afonso, Antoine Gessain.

**Writing – review & editing:** Olivier Cassar, Emilie Deruelle, Philippe Vicente Afonso, Antoine Gessain.

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
