## [Decision Letter · Decision Letter 0]

15 Mar 2024

Dear Dr. Cassar,

Thank you very much for submitting your manuscript "Southern African Origin for HTLV-1 in Romania" for consideration at PLOS Neglected Tropical Diseases. As with all papers reviewed by the journal, your manuscript was reviewed by members of the editorial board and by several independent reviewers. In light of the reviews (below this email), we would like to invite the resubmission of a significantly-revised version that takes into account the reviewers' comments. 

We cannot make any decision about publication until we have seen the revised manuscript and your response to the reviewers' comments. Your revised manuscript is also likely to be sent to reviewers for further evaluation.

Sincerely,

Adly M.M. Abd-Alla, Prof asso.

Academic Editor

Andrea Marzi

Section Editor

Reviewer's Responses to Questions

**Key Review Criteria Required for Acceptance?**

**Methods**

-Are the objectives of the study clearly articulated with a clear testable hypothesis stated?

-Is the study design appropriate to address the stated objectives?

-Is the population clearly described and appropriate for the hypothesis being tested?

-Is the sample size sufficient to ensure adequate power to address the hypothesis being tested?

-Were correct statistical analysis used to support conclusions?

-Are there concerns about ethical or regulatory requirements being met?

Reviewer #1: (No Response)

Reviewer #2: The main method of this study consists in the molecular analysis of these new HTLV sequences. Therefore, it is essential to include in methods how the sequences were obtained, and all phylogenetic analysis done. This information is currently spread in results session. Authors need to include it in methods.

First statement of results belongs to methods. And authors need to expand briefly on the method used (reference 12). 

“DNA samples were obtained from peripheral blood buffy coats. The LTR sequence was obtained for the 11 samples, as previously described (12)”

Figure 1 and 2 legends also contain important information that needs to be in methods.

The following statement in results also should be in methods.

“The DNA samples were subjected to four series of PCR (F1-F4) using both a high-fidelity hot-start DNA polymerase and primers designed from highly conserved regions for the major HTLV-1 subtypes. In brief, we obtained four different HTLV-1 proviral genomic regions, each about 2.5 kb in length: F1, LTR-Gag; F2, pro-pol; F3, pol-env and F4, tax-LTR. PCR products were then directly sequenced on both strands using a set of 16 specific primers (12). After alignment of sense and antisense sequences using the Clustal W algorithm (Mac Vector 18.6.1 software, Oxford molecular), comparison of each generated segment was implemented to derive a consensus sequence.”

Minor points:

- How many patients were tested for HBV?

- How many mothers were tested?

- Include in methods information about transfusion (the first time this is mentioned is in discussion). Was it done in Romania?

- If available, it would be interesting to know how long they have been out of Romania.

Reviewer #3: There appear no problems in the objectives, methods and interpretation of the results.

Ethical or regulatory requirements are properly met.

**Results**

-Does the analysis presented match the analysis plan?

-Are the results clearly and completely presented?

-Are the figures (Tables, Images) of sufficient quality for clarity?

Reviewer #1: (No Response)

Reviewer #2: Results

Authors says that all sequences except ROU7 belong to a-TC-Mozambique – Southern Africa A subgroup. What about the outlier (ROU7) ?

Authors did whole genome sequence of 10/11 sequences, please inform the reader which sample was missing. 

Authors say that stop mutations are quite frequent in ATL. This needs to go to discussion, and a reference is needed.

Authors say that 9/10 Romanian strain form a specific “Romania” clade. Please add in the text which sample is the exception. 

The last paragraph of result can be moved to discussion.

Suggest replacing “They” at the beginning of the second paragraph of results for something more specific Newly characterised sequences…

Reviewer #3: Figure 1 and 2: The figures are blurred and the texts in the figures unreadable.

The analysis itself is appropriate and the results are described clearly.

**Conclusions**

-Are the conclusions supported by the data presented?

-Are the limitations of analysis clearly described?

-Do the authors discuss how these data can be helpful to advance our understanding of the topic under study?

-Is public health relevance addressed?

Reviewer #1: (No Response)

Reviewer #2: Authors need to include a conclusion session.

Reviewer #3: 1. Page 14: I understand that all the viruses identified in Romania seem to originate from Mozambique, but there is little discussion on the background and reasons for this. Is this because of immigrants? Also, given that multiple individuals have been infected, shouldn't there be a discussion of how and when the virus entered the country and how it spread within the country? 

2. Page 14 and figure 2: The possible origin of the remaining one specimen should be discussed

3. Page 14, lines 11-13: Since many cases of ATL from horizontal infection have been confirmed in Japan, should the wording be weakened?

**Editorial and Data Presentation Modifications?**

Reviewer #1: (No Response)

Reviewer #2: (No Response)

Reviewer #3: As mentioned above, the figures in the PDF are very poor to understand what are described. This may be due to the problems in making PDF, however, the clear figures need to be provided to the reviewers.

**Summary and General Comments**

Reviewer #1: (No Response)

Reviewer #2: This manuscript brings novel and interesting data that sheds light on HTLV-1 spread to Romania, a high endemic country in Europe. The molecular analysis is done at high standard and shows convincing evidence of a recent African origin. 

However, some methods are currently described in result session or Figure legends. Discussion can also be expanded.

Authors could include discussion about current situation of HTLV response in Romania. Are they testing Blood donors? Are they testing organ donors? They can also discuss strategies that can limit the spread of this virus in Romania (as they suggested this should be done).

Another interesting point to discuss is how to limit spread to other countries. Recent blood donor data indicates high prevalence of HTLV in Moldova and brings concern about other countries within the region where no HTLV prevalence data is available. Authors have extensive experience on HTLV epidemiology and could expand on this topic. Another question that this study brings is if the origin of HTLV in Moldova is the same as they have revealed for Romania.

Reviewer #3: It is significant that the origin of HTLV-1 in Romania has been clarified.

However, there appears some room for improvement in interpretation and discussion of the data、in addition to the poor quality of the figures.

PLOS authors have the option to publish the peer review history of their article (what does this mean?). If published, this will include your full peer review and any attached files.

Reviewer #1: No

Reviewer #2: Yes: Carolina Rosadas

Reviewer #3: Yes: Toshiki Watanabe
---

## [Decision Letter · Decision Letter 1]

3 Jul 2024

Dear Dr. Cassar,

We are pleased to inform you that your manuscript 'Southern African Origin of HTLV-1 in Romania' has been provisionally accepted for publication in PLOS Neglected Tropical Diseases.

Best regards,

Adly M.M. Abd-Alla, Prof asso.

Academic Editor

Andrea Marzi

Section Editor

Reviewer's Responses to Questions

**Key Review Criteria Required for Acceptance?**

**Methods**

-Are the objectives of the study clearly articulated with a clear testable hypothesis stated?

-Is the study design appropriate to address the stated objectives?

-Is the population clearly described and appropriate for the hypothesis being tested?

-Is the sample size sufficient to ensure adequate power to address the hypothesis being tested?

-Were correct statistical analysis used to support conclusions?

-Are there concerns about ethical or regulatory requirements being met?

Reviewer #2: NA

Reviewer #3: The description of the method has been improved significantly with revisions suggested by the reviewers. Thus, readers can understand them clearly.

**Results**

-Does the analysis presented match the analysis plan?

-Are the results clearly and completely presented?

-Are the figures (Tables, Images) of sufficient quality for clarity?

Reviewer #2: NA

Reviewer #3: Responding to the reviewers’ comments, revised descriptions have been improved so that the readers can clearly understand the information.

**Conclusions**

-Are the conclusions supported by the data presented?

-Are the limitations of analysis clearly described?

-Do the authors discuss how these data can be helpful to advance our understanding of the topic under study?

-Is public health relevance addressed?

Reviewer #2: NA

Reviewer #3: The manuscript does not have an independent section of conclusions; however, interpretation and the significance of the results is presented in the discussion section, which appear to be reasonable and understandable to the readers.

**Editorial and Data Presentation Modifications?**

Reviewer #2: (No Response)

Reviewer #3: They are improved in the revised manuscript.

**Summary and General Comments**

Reviewer #2: I appreciate the changes made by authors, I believe they answer all questions appropriately and I am happy with the revised version of the manuscript.

Reviewer #3: The abstract section provides the summary and comments, which are reasonable based on the results obtained

PLOS authors have the option to publish the peer review history of their article (what does this mean?). If published, this will include your full peer review and any attached files.

Reviewer #2: **Yes: **Carolina Rosadas

Reviewer #3: No

---

## [Editor Report · Acceptance letter]

23 Jul 2024

Dear Dr. Cassar,

We are delighted to inform you that your manuscript, "Southern African Origin of HTLV-1 in Romania," has been formally accepted for publication in PLOS Neglected Tropical Diseases.

Best regards,

Shaden Kamhawi

co-Editor-in-Chief

Paul Brindley

co-Editor-in-Chief
